

**Nighttime O($^1$D) distributions in the mesopause region derived from**
**SABER data**
Mikhail Yu. Kulikov[1], and Mikhail V. Belikovich[1]
[1]Institute of Applied Physics of the Russian Academy of Sciences, 46 Ulyanov Str., 603950 Nizhny Novgorod, Russia
*Correspondence to*: Mikhail Yu. Kulikov (mikhail_kulikov@mail.ru)
**Abstract.** In this study, the new source of O($^1$D) in the mesopause region proposed by Kalogerakis (2019) is applied to
SABER data to estimate the nighttime O($^1$D) distributions for the years 2003-2005. It is found that O($^1$D) evolutions in these
years are very similar to each other. Depending on the month, monthly averaged O($^1$D) distributions may have a pronounced
maximum (up to 80-110 cm$^{-3}$ in January and July) localized in height (at ~97±4 km) and latitude (at ~52±4°S or ~52±4°N).
The nightly averaged O($^1$D) concentrations may reach ~300 cm$^{-3}$. The obtained results are useful data set for subsequent
estimation of nighttime O($^1$D) influence on chemistry of the mesopause region.
**1 Introduction**
Daytime O($^1$D) is considered to be one of the important chemical minor species of the stratosphere, mesosphere and
thermosphere, as it plays a significant role in the chemistry, and the radiative and thermal balance of this region (Brasseur &
Solomon, 2005). First of all, formed by photolysis of $O_2$ and $O_3$, O($^1$D) is a mediator involved in the transformation of
absorbed solar radiation energy into the heating of this region and, in particular, excitation of $N_2(v)$ and $CO_2(v)$ (Harris &
Adams, 1983; Panka et al., 2017). Also, O($^1$D) atoms participate in the reactions of destruction of long-lived greenhouse
gases (Baasandorj et al., 2012), $CH_4$ oxidation, and $HO_x$ and $NO_x$ production, for example:
O($^1$D)+$N_2$O $\rightarrow$ 2NO
O($^1$D)+$H_2$O $\rightarrow$ 2OH
O($^1$D)+$H_2$ $\rightarrow$ H+OH
O($^1$D)+$CH_4$ $\rightarrow$ $CH_3$+OH
O($^1$D)+$CH_4$ $\rightarrow$ $H_2$+$CH_2$O
Moreover, the red line emission from O($^1$D) atoms is one of the most important airglow phenomenon which are used as a
diagnostic of the ionosphere, for example, to monitor the electron density and neutral winds in the F region (Shepherd et al.,
2019). Therefore, many papers and experimental campaigns are devoted to measurements of features of $O_3$ photolysis to
O($^1$D) (Taniguchi et al., 2003; Hofzumahaus et al., 2004).
Until recently, it was believed that the above mentioned processes stopped at night due to absent of constant source of O($^1$D)
in this time and extremely low (less than 1 s) life time of this component. In principle, O($^1$D) can be generated in sprite halos
but for a low duration of 1 ms (Hiraki et al., 2004). In this year, based on laboratory experiments, Kalogerakis (2019)
highlighted a previously unrecognized source of nighttime O($^1$D) and $O_2$ A-band emission in the mesopause region via
process:
OH($v{\geq}5$) +O($^3$P) $\rightarrow$ OH($0{\leq}v^{'}{\leq}v$-5) + O($^1$D),                    (1)
that is multiquantum quenching of high excited states of OH by collisions with atomic oxygen in ground state. Taking into
account the major way of this process (Kalogerakis et al., 2016):
OH(9) +O($^3$P) $\rightarrow$ OH(3) + O($^1$D),                    (2)
Kalogerakis (2019) showed that a new model of $O_2$ A-band well described (qualitatively and quantitatively) the results of
early nighttime rocket measurements of volume emission rate profiles of this airglow. Thus, he proved that the process (1)
really took place in mesopause region and the way (2) was the major source of O($^1$D).



In this study, the new source of O($^1$D) in the mesopause region proposed by Kalogerakis (2019) is applied to SABER data to
estimate the O($^1$D) nighttime distributions for the years 2003-2005.

**2 O($^1$D) model and method of derivation from SABER Data**

Following Kalogerakis (2019), the nighttime balance of OH(9) and O($^1$D) concentrations in the mesopause region is
determined by processes summarized in Table 1. Due to low values of chemical lifetimes (less than 1 s), these components
can be considered in chemical equilibrium:
$$OH(9) = \frac{y_9 \cdot k_1 \cdot H \cdot O_3}{k_2 \cdot O_2 + k_3 \cdot N_2 + k_4 \cdot O + k_5},$$ (3)
$$O(^1D) = \frac{y_1 \cdot k_4 \cdot OH(9) \cdot O}{k_6 \cdot O_2 + k_7 \cdot N_2 + k_8} = \frac{y_1 \cdot y_9 \cdot k_1 \cdot k_4 \cdot O \cdot H \cdot O_3}{(k_2 \cdot O_2 + k_3 \cdot N_2 + k_4 \cdot O + k_5) \cdot (k_6 \cdot O_2 + k_7 \cdot N_2 + k_8)},$$ (4)
where $k_i$ are the corresponding process rate coefficients. Thus, local O($^1$D) concentration is defined by the values of
temperature (T), and concentrations of M, O$_3$, O, and H. We suggest getting this information from satellite-based
observations.
Mlynczak et al. (2013, 2014) proposed the method of nighttime O and H derivation in the range of 0.01–0.0001 hPa
(approximately 80–105 km) from simultaneous measurements of temperature, ozone (using ozone emission at 9.6 μm) and
OH(9–7) and OH(8–6) band emissions by the SABER (Sounding of the Atmosphere using Broadband Emission
Radiometry) instrument onboard the TIMED (Thermosphere Ionosphere Mesosphere Energetics and Dynamics) satellite.
The method used two assumptions: the chemical equilibrium condition for nighttime ozone, and the model of OH(9–7) and
OH(8–6) emissions. Recently (Mlynczak et al. (2018), the parameters of the model was corrected. So now, O distributions
derived from SABER data are in good consistent with O distributions obtained from SCIAMACHY green-line and OH
nightglow measurements (Zhu & Kaufmann, 2019). In this work, we derive the local values of O and H from SABER data
and apply all sets of data (T, concentrations of M, O$_3$, O, and H) to retrieve the local concentrations of O($^1$D) with the use of
eq. (4). At this, we use also the analytical criterion (Kulikov et al., 2018) that allows the localization of the lower boundary
of nighttime ozone chemical equilibrium (Kulikov et al., 2019) with the use of SABER data.

**3 O($^1$D) nighttime distributions**

We use the version 2.0 of the SABER data product (Level2A) for the simultaneously measured O$_3$, volume emission rate of
OH from the $v = 9$ and $v = 8$ states and temperature profiles within the 0.01–0.0001 hPa pressure ($p$) interval (approximately
80–105 km in 2003-2005. We take only nighttime data when the solar zenith angle χ > 95°. Appling the mentioned criterion
for each set of simultaneously measured profiles, we find the local position (the pressure level $p_{eq}$) of the boundary of
nighttime ozone chemical equilibrium. Thus, we take into account only the upper part of each SABER profile corresponding
$p \geq p_{eq}$. The range of latitudes covered by the satellite trajectory in a month was divided into 20 bins ~ (5-8)° each. 1500-
3000 single profiles of O($^1$D) concentration fall into one bin during a month of SABER observations (or 50-100 profiles per
a one night). For each bin we calculate monthly and nightly averaged zonal mean $< O(^1D) >$ distributions (hereafter, the
angle brackets are used to denote timely and spatially averaged values).
Monthly averaged $< O(^1D) >$ distributions in corresponding month of 2003-2005 are shown in Figs. 1–3. Firstly, it can be
noted that O($^1$D) evolutions in these years are very similar to each other. Secondly, many features of O($^1$D) in the southern
hemisphere are repeated in the northern hemisphere with a shift of 6 months. In particular, O($^1$D) concentration distributions
in January-February and November-December have a pronounced maximum (up to 80 cm$^{-3}$ in January) localized in height
(at ~97±4 km) and latitude (at ~52±4°S). In May-August, the distributions have similar maximum (up to 110 cm$^{-3}$ in July)
localized at ~98±3 km and ~52±4°N. In other months (March-April and September-November), one can see transitional
O($^1$D) distributions with several maxima but their values don't exceed (30-35) cm$^{-3}$. Figs. 4–5 show the nightly averaged


$< O(^1D) >$ vertical distributions at (48-54)°N and (48-54)°S as a function of day of year. Examples of these profiles are
presented in Fig. 6. One can see that local value of $< O(^1D) >$ at ~(97-98) km may reach ~300 cm$^{-3}$ in both hemispheres.
The uncertainty of local O($^1$D) concentration is defined mainly by local uncertainty of O derivation. Taking into account the
O uncertainty profile presented in Mlynczak et al. (2013), we estimate that uncertainty of local O($^1$D) varies in the range of
(30-40)% depending on the pressure level. Due to averaging, the uncertainty of nightly averaged O($^1$D) shown in Fig. 6 is
estimated to be less than 6%.
**4 Discussion and Conclusion**
According to different early papers (Nicolet, 1959; Ghosh & Gupta, 1970; Shimazaki & Laird, 1970; Harris & Adams,
1983), daytime O($^1$D) concentrations at 90-100 km varied in the range of ($10^2$-$10^4$) cm$^{-3}$. Brasseur & Solomon (2005)
published the table (see Table A.6.2.c) where daytime O($^1$D) changed from 70 cm$^{-3}$ at 90 km to 140 cm$^{-3}$ at 100 km. The
presented results show that monthly and nightly mean nighttime O($^1$D) concentrations at these altitudes can reach 100 cm$^{-3}$
and 300 cm$^{-3}$, respectively. Thus, nighttime concentrations of O($^1$D) are comparable with daytime concentrations of this
component and, in principle, can impact noticeably the chemistry and thermal balance of the mesopause region. More
detailed analyze of this impact should be carried out with the use of the global 3D chemical transport model of the
mesosphere – lower thermosphere.
**Data availability.** The SABER data used in this study can be downloaded from ftp://saber.gats-
inc.com/Version2_0/Level2A/.        The        presented        data        can        be        downloaded        from
http://www.iapras.ru/english/structure/dep_240/dep_240.html.
**Author contributions.** Both authors contributed equally to this paper.
**Competing interests.** The authors declare that they have no conflict of interest.
**Acknowledgments.** The authors are grateful the SABER team for data availability.

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






| | Process | Rate coefficient |
|---|---|---|
| 1 | $H + O_3 \rightarrow O_2 + OH(v)$ | $1.4 \cdot 10^{-10} \cdot \exp(-470/T)$ cm$^3$ s$^{-1}$ OH(9) yield is $y_9 = 0.47$. |
| 2 | $OH(9) + O_2 \rightarrow$ products | $1.15 \cdot 10^{-11} \cdot \exp(195/T)$ cm$^3$ s$^{-1}$ |
| 3 | $OH(9) + N_2 \rightarrow$ products | $5.03 \cdot 10^{-13} \cdot \exp(100/T)$ cm$^3$ s$^{-1}$ |
| 4 | $OH(9) + O \rightarrow$ products | $6.2 \cdot 10^{-10} \cdot \exp(-135/T)$ cm$^3$ s$^{-1}$ O($^1$D) yield is $y_1 = 5/6.2$ |
| 5 | radiative decay of OH(9) | $173$ s$^{-1}$ |
| 6 | $O(^1D) + O_2 \rightarrow O + O_2$ | $3.3 \cdot 10^{-11} \cdot \exp(55/T)$ cm$^3$ s$^{-1}$ |
| 7 | $O(^1D) + N_2 \rightarrow O + N_2$ | $2.15 \cdot 10^{-11} \cdot \exp(110/T)$ cm$^3$ s$^{-1}$ |
| 8 | radiative decay of O($^1$D) | $0.009$ s$^{-1}$ |

**Table 1. List of processes with corresponding rate coefficients from Kalogerakis et al. (2011), (2016), Sharma et al.**
**(2015) and Burkholder et al. (2015).**





**Figures**

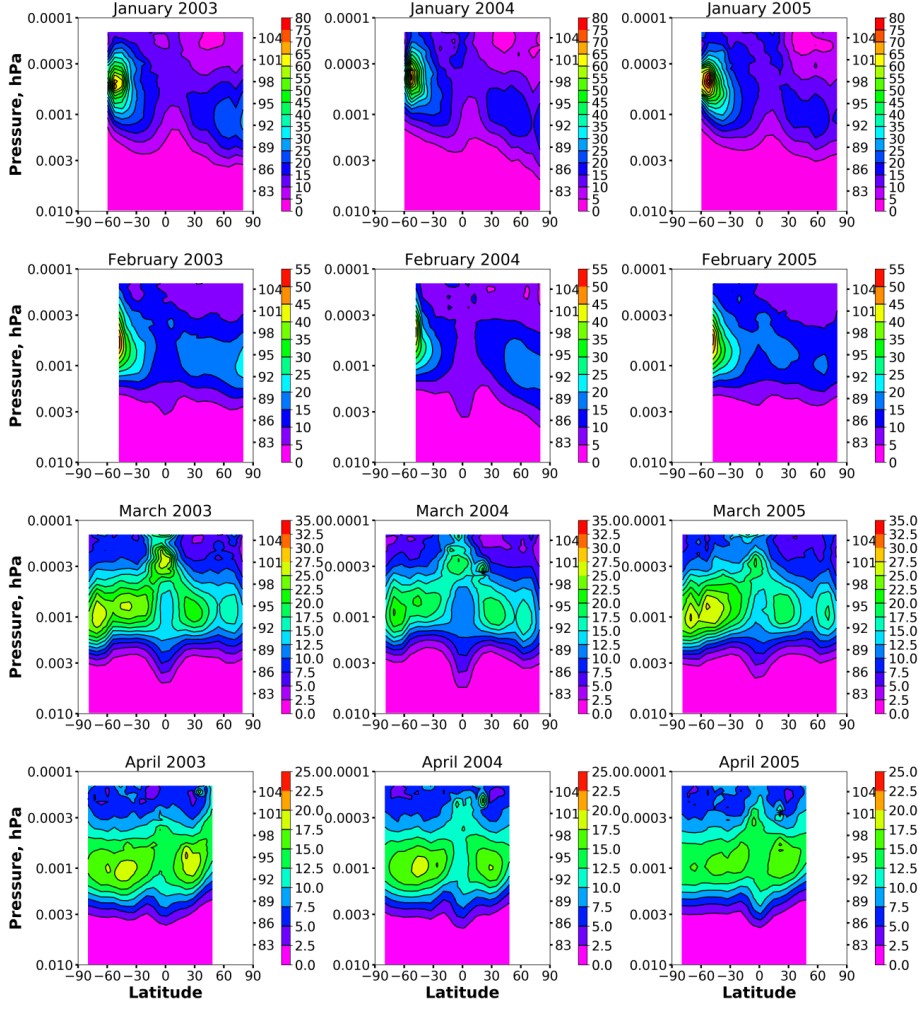



**Figure 1. Monthly averaged O($^1$D) concentration (in cm$^{-3}$) in different months of 2003-2005.**






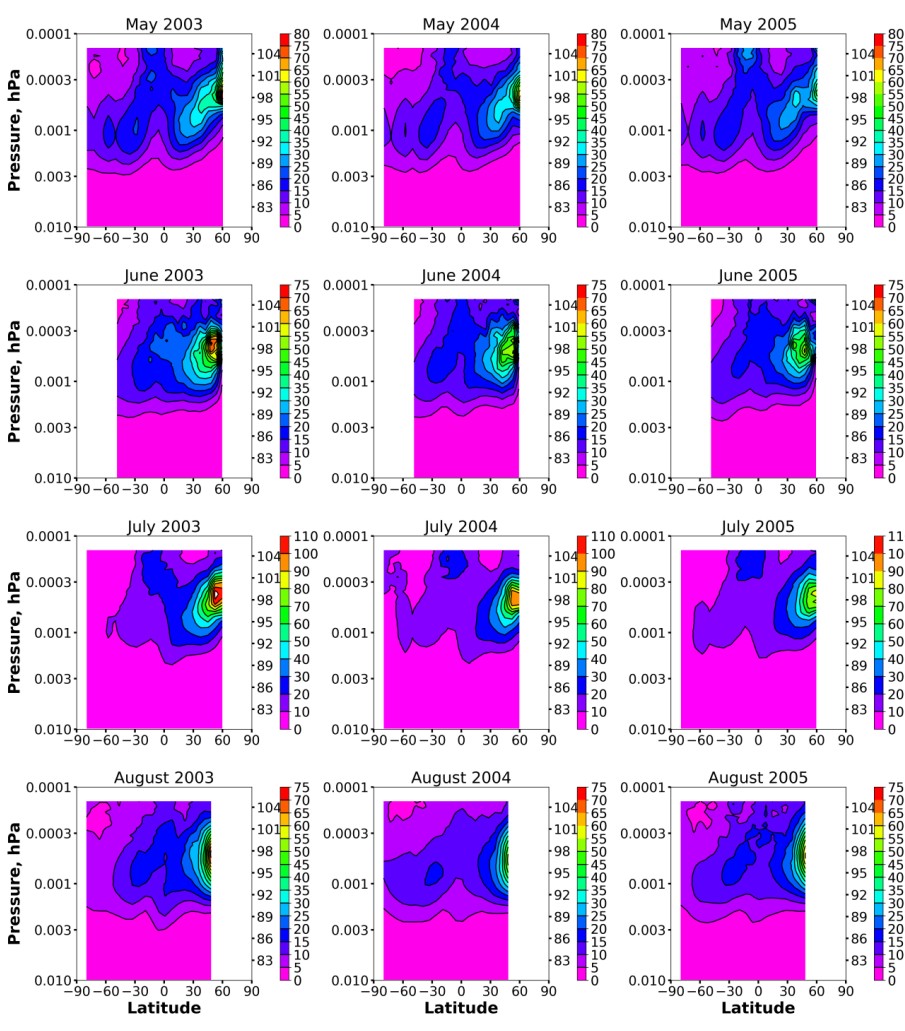


**Figure 2. Monthly averaged O($^1$D) concentration (in cm$^{-3}$) in different months of 2003-2005.**







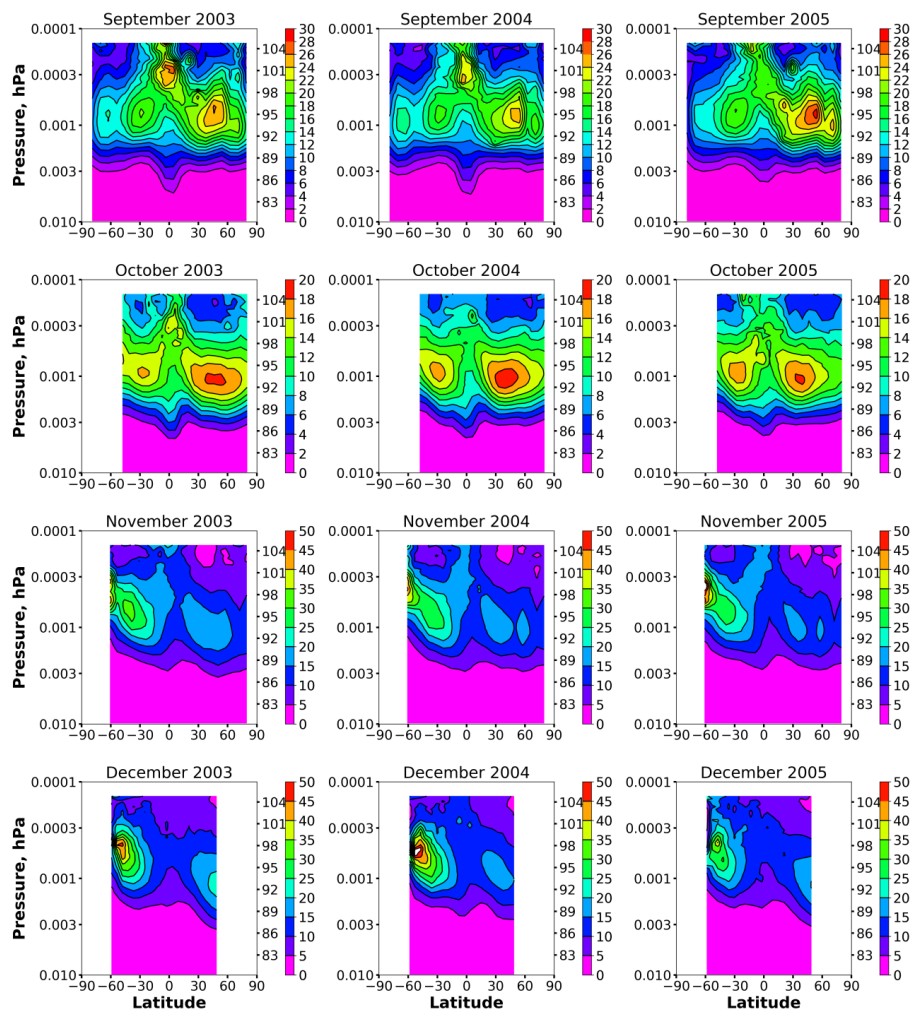


**Figure 3. Monthly averaged O($^1$D) concentration (in cm$^{-3}$) in different months of 2003-2005.**







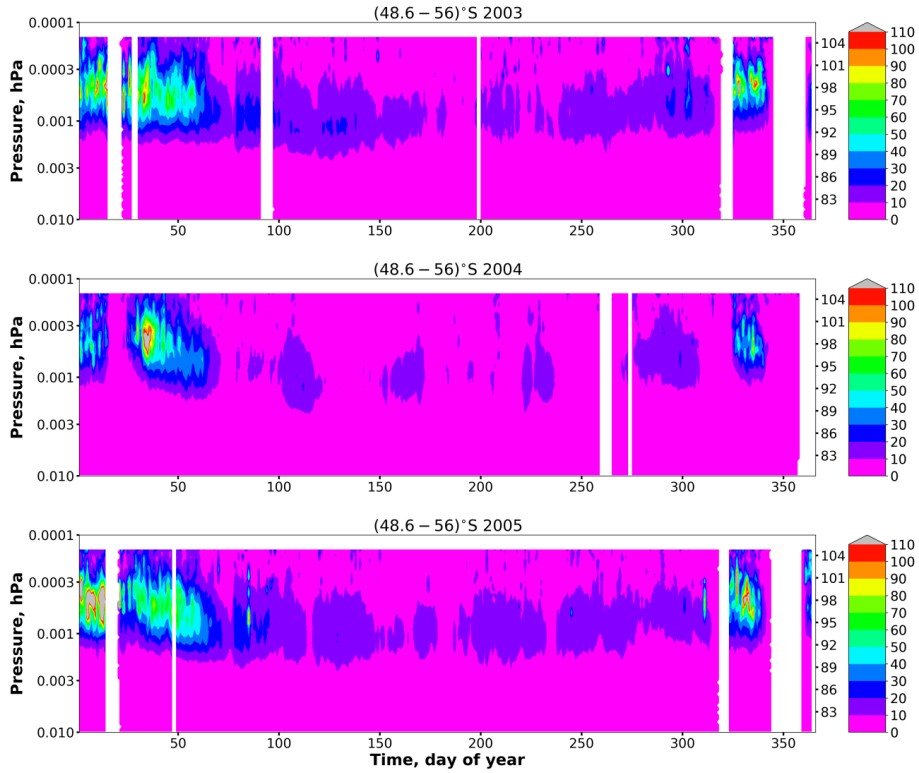

**Figure 4. The nightly averaged $< O(^1D) >$ vertical distributions at (48-54)°S as a function of day of year. The values of O($^1$D)**
**concentration greater than 110 cm$^{-3}$ are grayed out.**






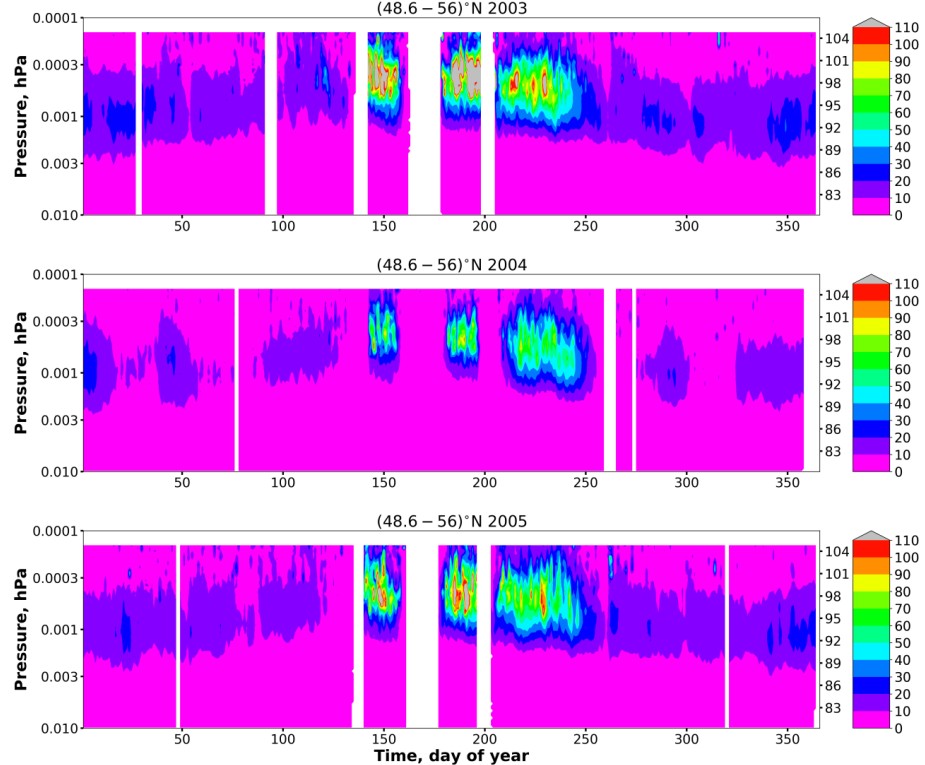

**Figure 5. The nightly averaged $< O(^1D) >$ vertical distributions at (48-54)°N as a function of day of year. The values of O($^1$D)**
**concentration greater than 110 cm$^{-3}$ are grayed out.**




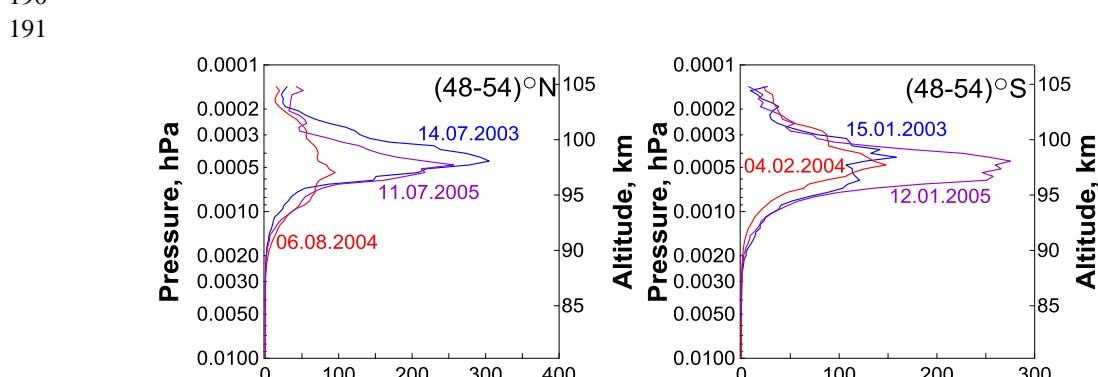

**Figure 6. Examples of nightly averaged** $< O(^1D) >$ **profiles vertical distributions at (48-54)°N (left) and (48-54)°S (right).**