# Peer review of "Nighttime O(1D) distributions in the mesopause region derived from"

_Annales Geophysicae, 2019_

## Referee Comment (RC1) · Anonymous Referee #1 · 25 Dec 2019

The manuscript applies a new source of O(1D) proposed by Kalogerakis (2019) to data from SABER to estimate the nighttime O(1D) population distributions for the years 2003-2005. The motivation of the study is to provide information for subsequent evaluation of nighttime O(1D) influence on the chemistry of the mesopause region. The manuscript reports that depending on the time of year, monthly averaged O(1D) distributions may have a pronounced maximum localized in height and latitude. The nightly averaged O(1D) concentrations may reach number densities as high as 300 cm-3.

The strength of the manuscript is that it is a clearly written paper related to a topic currently debated in the literature and its motivation to provide helpful information to

better understand mesopause chemistry is justified. The weakness of the manuscript is that the calculation of [O(1D)] has major flaws and inconsistencies.

The fact that there are no measurements of [O1D] near the mesopause means there is no direct comparison between results from observations and calculated estimates. As a consequence, the details of the calculation of [O(1D)] must be considered very carefully.

There are several problems with the calculations reported in this manuscript:

(a) The SABER profiles for oxygen atoms used in the calculations are inconsistent with the new source of O(1D) and not appropriate for this type of calculation. These oxygen atom profiles come from a model which assumes 100% single-quantum relaxation by oxygen atoms from OH(9) to OH(8) to get the global energy budget near balance (Mlynczak et al., 2018). The yield of the new mechanism from OH(9) to OH(8) is less than 1.2/6.2 or ∼20% according to Table 1 of the manuscript. Applying the results of the SABER single-quantum model to calculate [O(1D)] is in contradiction with the new multi-quantum O(1D) source. In addition, using the single-quantum approach described in Mlynczak et al. (2018) was recently shown to give inconsistent results with SCIAMACHY data (Fytterer et al., ACP, 2019).

(b) The SABER inputs for hydrogen atoms used in the calculations suffer from the same problem as discussed in (a) above. The best-fit results for [H] calculated by the multi-quantum model of Fytterer et al. (2019) are 50% larger than the single-quantum SABER model. The hydrogen atom population distributions are also strongly affected from uncertainty in ozone, to be discussed next.

(c) The SABER inputs for ozone are not well constrained. In order to get the global energy budget near balance, Mlynczak et al. (2018) made an arbitrary adjustment reducing the daytime ozone values between 65 and 100 km by 25%, and also considered the possibility of nighttime ozone being too high. Both [H] and [O3] directly affect the calculation of [O(1D)] and any systematic errors are multiplied.

(d) There are at least three very different values in the recent literature for the rate coefficient of OH(9) + O (Kalogerakis, 2019; Fytterer et al., 2019; Zhu and Kaufmann, GRL, 2018). The choice of this rate coefficient directly affects the calculations. The manuscript ignores these possibilities and their effect on the estimated [O(1D)]. For example, at a mesopause temperature of 190 K, the manuscript adopts a rate coefficient for OH(9) + O of 3.05 x 10(-10) cm3s-1, whereas Zhu and Kaufmann (2018) determined a best-fit value of 2.3 x 10(-10) cm3s-1. This choice implies ∼33% larger [O(1D)] for the calculation reported in the manuscript.

In contrast to the assertion of the manuscript, the SCIAMACHY data and latest SABER atomic oxygen data reveal significant systematic differences at all latitudes and seasons (Zhu and Kaufmann, 2018). Below 87 km, the SABER atomic oxygen dataset is 40% lower on average than SCIAMACHY. This may be attributed to a very unrealistic, rate coefficient for quenching of OH(8) by O2 used in the SABER single-quantum model (approximately a factor of 50 smaller than the corresponding quenching rate of OH(9) by O2). Above 90 km, the difference between the two datasets is reversed and the SABER atomic oxygen is 10-30% higher than SCIAMACHY. Additionally, the SABER photochemical model for ozone does not take into account the loss of ozone through reaction with atomic oxygen, which affects retrieved atomic oxygen on the order of 30% at atomic oxygen peak altitudes. All these differences summarized in the discussion above directly propagate into the calculation of [O(1D)], which cannot be directly validated by observations.

In conclusion, several important parameters used for the calculation of [O(1D)] in this manuscript are flawed and inconsistent with the multi-quantum source of O(1D), and therefore lead to inaccurate results and large systematic errors.

Other Minor Comments ———————————— Line 37 "…A-band is well…"

Line 57 Delete "good" — it is redundant

Line 92 "…detailed analysis…"

Line 99 "…grateful to the…"

---

## Author Comment (AC1) · 24 Jan 2020

Below Referee's comments are marked by red.

(a) The SABER profiles for oxygen atoms used in the calculations are inconsistent with the new source of O($^1$D) and not appropriate for this type of calculation. These oxygen atom profiles come from a model which assumes 100% single-quantum relaxation by oxygen atoms from OH(9) to OH(8) to get the global energy budget near balance (Mlynczak et al., 2018). The yield of the new mechanism from OH(9) to OH(8) is less than 1.2/6.2 or ~20% according to Table 1 of the manuscript. Applying the results of the SABER single-quantum model to calculate [O($^1$D)] is in contradiction with the new multi-quantum O($^1$D) source. In addition, using the single-quantum approach described in Mlynczak et al. (2018) was recently shown to give inconsistent results with SCIAMACHY data (Fytterer et al., ACP, 2019).

(b) The SABER inputs for hydrogen atoms used in the calculations suffer from the same problem as discussed in (a) above. The best-fit results for [H] calculated by the multi-quantum model of Fytterer et al. (2019) are 50% larger than the single-quantum SABER model. The hydrogen atom population distributions are also strongly affected from uncertainty in ozone, to be discussed next.

We agree with both comments. As it was above mentioned, in the revised manuscript we used the new OH(v) model from Fytterer et al. (ACP, 2019). Their 'the best-fit model' includes all commonly used production and loss processes of OH(v) (see Table 1 in the article), in particular, single- and multi-quantum relaxation by oxygen atoms and the new multi-quantum O(1D) source via OH($v{\geq}5$) +O($^3$P) $\rightarrow$ OH($0{\leq}v^{'}{\leq}v$-5) + O($^1$D). Fytterer et al. adjusted some parameters of the model (in particular, branching ratios of quenching OH(v)+O2 and rate coefficients of OH($v{\geq}5$) +O($^3$P) $\rightarrow$ OH($0{\leq}v^{'}{\leq}v$-5) + O($^1$D)) with the use of volume emission rate profiles in four different wavelength measured by SABER and SCIAMACHY.

In the revised manuscript, O($^3$P), OH(v=3-9), and O($^1$D) are calculated using the simultaneous measurements of O$_3$ (9.6 μm), volume emission rate of (9-7) and (8-6) OH transitions (VER_2) and temperature (T). It is done with the use of following system of equations:

$$\begin{cases} P_{OH} = k_3 \cdot H \cdot O_3 = k_1 \cdot M \cdot O_2 \cdot O - k_2 \cdot O_3 \cdot O \\ VER\_2 = EC_{97} \cdot OH(9) + EC_{86} \cdot OH(8) \\ \quad OH(9,8) = F(P_{OH}, O, T, M) \end{cases} \tag{1}$$

where the first equation follows from the nighttime ozone chemical equilibrium assumption; $P_{OH}$ is the source of OH(v) via the reaction H + O$_3$ $\rightarrow$ O$_2$ + OH(v); $M$ is air concentration;

$k_{1-3}$ are rate coefficients of reactions O+O$_2$+M $\rightarrow$ O$_3$+M, O+O$_3$ $\rightarrow$ 2O$_2$, and H + O$_3$ $\rightarrow$ O$_2$ + OH($v$) correspondingly; $EC_{97}$ and $EC_{86}$ are Einstein coefficients for OH transitions (9-7) and (8-6) correspondingly; $OH(9,8)$ are known functions of $P_{OH}, O, T, M$ from Fytterer et al. model.

(c) The SABER inputs for ozone are not well constrained. In order to get the global energy budget near balance, Mlynczak et al. (2018) made an arbitrary adjustment reducing the daytime ozone values between 65 and 100 km by 25%, and also considered the possibility of nighttime ozone being too high. Both [H] and [O$_3$] directly affect the calculation of [O($^1$D)] and any systematic errors are multiplied.

It is undeniable that systematic errors of nighttime [O3] measurements directly affect our calculations. Currently, we do not observe the consensus in the literature about those errors. As we can see, Fytterer et al. (ACP, 2019) came to the different conclusion about SABER nighttime O$_3$ measurements. In particular, Fytterer et al. wrote (see p. 1846) «Recent comparisons between MIPAS O$_3$ and SABER O$_3$ derived at 9.6 μm were performed by López-Puertas et al. (2018). The authors showed that night-time O3 from SABER is slightly larger than night-time O$_3$ obtained from MIPAS in the altitude region 80–100 km over the Equator (their Figs. 8 and 10), but these differences are within the corresponding errors. Thus, at least to our knowledge, there is no conclusive evidence stating that SABER night-time O$_3$ is generally too large.»

(d) There are at least three very different values in the recent literature for the rate coefficient of OH(9) + O (Kalogerakis, 2019; Fytterer et al., 2019; Zhu and Kaufmann, GRL, 2018). The choice of this rate coefficient directly affects the calculations. The manuscript ignores these possibilities and their effect on the estimated [O($^1$D)]. For example, at a mesopause temperature of 190 K, the manuscript adopts a rate coefficient for OH(9) + O of 3.05 x 10(-10) cm3s-1, whereas Zhu and Kaufmann (2018) determined a best-fit value of 2.3 x 10(-10) cm3s-1. This choice implies 33% larger [O($^1$D)] for the calculation reported in the manuscript.

We agree with the comment. The OH($v$) model from Fytterer et al. (ACP, 2019) use the same (see Table 3) rate coefficient for O($^1$D) production due to OH(9) + O as Zhu and Kaufmann (2018). Moreover, Fytterer et al. determined branching ratios of OH($v$) +O($^3$P) $\rightarrow$ OH($v'$) + O($^1$D) based on the OH(6-2) VER, OH(5-3)+(4-2) VER, and OH(3-1) VER observations.

In contrast to the assertion of the manuscript, the SCIAMACHY data and latest SABER atomic oxygen data reveal significant systematic differences at all latitudes and seasons (Zhu and Kaufmann, 2018). Below 87 km, the SABER atomic oxygen dataset is 40% lower on average than SCIAMACHY. This may be attributed to a very unrealistic, rate coefficient for quenching of OH(8) by $O_2$ used in the SABER single-quantum model (approximately a factor of 50 smaller than the corresponding quenching rate of OH(9) by O2). Above 90 km, the difference between the two datasets is reversed and the SABER atomic oxygen is 10-30% higher than SCIAMACHY.

Additionally, the SABER photochemical model for ozone does not take into account the loss of ozone through reaction with atomic oxygen, which affects retrieved atomic oxygen on the order of 30% at atomic oxygen peak altitudes. All these differences summarized in the discussion above directly propagate into the calculation of $[O(^1D)]$, which cannot be directly validated by observations.

In conclusion, several important parameters used for the calculation of [O(1D)] in this manuscript are flawed and inconsistent with the multi-quantum source of O(1D), and therefore lead to inaccurate results and large systematic errors.

We agree with the comment. As it was above mentioned, we used more advanced OH(v) model in the revised manuscript which addressed to these concerns. Moreover, we applied the chemical equilibrium condition for nighttime ozone with taking into account the loss of ozone through reaction with atomic oxygen (see system (1) in this response).

Other Minor Comments

Line 37 "...A-band is well…"

Line 57 Delete "good" — it is redundant

Line 92 "…detailed analysis…"

Line 99 "…grateful to the…"

Corrected.